# The Enigma of Norbormide, a *Rattus*-Selective Toxicant

**DOI:** 10.3390/cells13090788

**Published:** 2024-05-06

**Authors:** Fabio Fusi, Simona Saponara, Margaret A. Brimble, David Rennison, Brian Hopkins, Sergio Bova

**Affiliations:** 1Department of Biotechnologies, Chemistry and Pharmacy, University of Siena, Via A. Moro 2, 53100 Siena, Italy; fabio.fusi@unisi.it; 2Department of Life Sciences, University of Siena, Via A. Moro 2, 53100 Siena, Italy; 3School of Chemical Sciences, University of Auckland, 23 Symonds Street, Auckland 1010, New Zealand; m.brimble@auckland.ac.nz (M.A.B.); d.rennison@auckland.ac.nz (D.R.); 4Manaaki-Whenua–Landcare Research, Canterbury Agriculture and Science Centre, 76 Gerald Street, Lincoln 7608, New Zealand; hopkinsb@landcareresearch.co.nz; 5Department of Pharmaceutical and Pharmacological Sciences, University of Padova, Via 8 Febbraio 2, 35122 Padova, Italy; sergio.bova@unipd.it

**Keywords:** norbormide, *Rattus*-selective toxicant, rodenticide, vasoconstriction

## Abstract

Norbormide (NRB) is a *Rattus*-selective toxicant, which was serendipitously discovered in 1964 and formerly marketed as an eco-friendly rodenticide that was deemed harmless to non-*Rattus* species. However, due to inconsistent efficacy and the emergence of second-generation anticoagulants, its usage declined, with registration lapsing in 2003. NRBs’ lethal action in rats entails irreversible vasoconstriction of peripheral arteries, likely inducing cardiac damage: however, the precise chain of events leading to fatality and the target organs involved remain elusive. This unique contractile effect is exclusive to rat arteries and is induced solely by the endo isomers of NRB, hinting at a specific receptor involvement. Understanding NRB’s mechanism of action is crucial for developing species-selective toxicants as alternatives to the broad-spectrum ones currently in use. Recent research efforts have focused on elucidating its cellular mechanisms and sites of action using novel NRB derivatives. The key findings are as follows: NRB selectively opens the rat mitochondrial permeability transition pore, which may be a factor that contributes to its lethal effect; it inhibits rat vascular K_ATP_ channels, which potentially controls its *Rattus*-selective vasoconstricting activity; and it possesses intracellular binding sites in both sensitive and insensitive cells, as revealed by fluorescent derivatives. These studies have led to the development of a prodrug with enhanced pharmacokinetic and toxicological profiles, which is currently undergoing registration as a novel efficacious eco-sustainable *Rattus*-selective toxicant. The NRB-fluorescent derivatives also show promise as non-toxic probes for intracellular organelle labelling. This review documents in more detail these developments and their implications.

## 1. Introduction: The History of a Mystery

### 1.1. The Discovery of Norbormide: A Historical Overview

Norbormide (NRB) is an acutely acting, *Rattus*-selective toxicant initially discovered during routine pharmacological screening at McNeil Laboratories (Fort Washington, PA, USA). Synthesised in pursuit of a novel anti-rheumatic drug, it proved ineffective. Subsequently investigated as an appetite suppressant, it demonstrated no toxic effects in house mice (*Mus musculus*) and cats (*Felis catus*) but revealed remarkable susceptibility in rats (*Rattus norvegicus* and *Rattus rattus*). This distinct susceptibility of rats, initially perceived as an error, was later confirmed through repeated experiments [1,2].

Further research led to its introduction as a species-selective rodenticide [3], marketed as Raticate^®^ and Shoxin^®^. Taste aversion and poor field efficacy, however, limited its effectiveness [4], and NRBs’ use dwindled in the 1970s. This decline was accelerated by the globally growing popularity of using anticoagulant toxicants, such as warfarin, for rodent control in both crop protection and conservation [5].

To date, the toxicants most commonly used to control rodent populations are the environmentally persistent anticoagulants that possess a broad spectrum of action and have the potential to kill many non-target species [6,7].

The first-generation anticoagulants required multiple consecutive ingestions to be lethal, while the second-generation anticoagulants, being more toxic, require only a single dose to induce the lethal effect. Both first- and second-generation anticoagulants share common flaws, as primary exposure to them is dangerous to both humans (especially children) and non-target animals, and they present severe secondary toxicity issues, particularly to predatory and/or scavenger species that prey upon poisoned rodents [8,9]. Many raptor species are highly sensitive to anticoagulant poisoning due to limited detoxifying capability [9,10]: the increasing death rate of these predators not only disrupts the ecosystem balance but has also seriously contributed to the extinction of many raptor species over the last 50 years.

The risks associated with the excessive use of anticoagulants are not limited to wildlife. Recent analyses, in fact, have found residues of anticoagulants in treated and untreated wastewater, sediments of water courses, suspended particulate matter, as well as in the liver of freshwater fish. The presence of anticoagulants in aquatic wildlife, most likely arising from their bioaccumulation in the food chain, may have critical consequences for the same wildlife, domestic animals, and human health [10,11]. As a result, governments are being pressured to restrict anticoagulant use in favour of novel, more pest-selective, eco-sustainable toxicants.

Consequently, due to its unique species-selectivity, renewed interest in the further development of NRB as a commercial rodenticide product has grown: this feature, if rationally exploited, would allow the safe control of many of the key invasive rat pest species, without endangering humans, other animals, and the environment [7].

Recent studies have suggested that NRBs’ poor performance in field conditions was probably the result of poor palatability due to an acute onset of symptoms leading to reduced bait consumption, resulting in sub-lethal dosing and the development of the phenomenon called *bait shyness* [12,13]. To overcome bait shyness at the time, Greaves et al. [14] used gelatine microencapsulation in an attempt to delay the onset of symptoms; however, in vivo, they found it did not alter the toxicity or the speed of action, and so the acute onset of symptoms, and the resulting bait shyness and low efficacy, remained a challenge.

A later attempt by Nadian and Lindblom [15], using gelatine microspheres over-coated with either shellac resin or an equal mixture of shellac and Eudragit RS^®^, substantially achieved the desired delay in the delivery of a lethal dose of NRB, though its release through mastication in feeding trials remained an issue. More recent attempts to improve NRB palatability include the development of either prodrug derivatives, which require cleavage to the active form after consumption and absorption as a means of delaying the onset of symptoms [16], or novel formulations allowing the preliminary consumption necessary to deliver a lethal dose [17]. Both products tested in the field attained excellent results and are progressing through product development and registration in New Zealand and globally.

### 1.2. Norbormide Physical Properties

Norbormide, 5-(α-hydroxy-α-2-pyridylbenzyl)-7-(α-2-pyridylbenzylidene)-5-norbornene-2,3-dicarboximide (CAS 991-42-4) (Figure 1), molecular weight of 511.55 g/mol, is a colourless to off-white solid with a melting point of 190–198 °C. Its solubility features are as follows: 60 mg/L in water, 14 mg/L in ethanol, >150 mg/L in chloroform, 1 mg/L in diethyl ether, 29 mg/L in 0.1 N HCl (all at 30 °C) [18], and >26 mg/mL in dimethylformamide and dimethyl sulfoxide [19]. As a dry solid, it is stable at room temperature, remaining unchanged for many years, and it is hydrolysed in an alkaline medium [19].

### 1.3. Norbormide Chemical Synthesis and Stereoisomerism

A review of literature synthetic protocols describing the preparation of NRB unequivocally revealed a two-step process—proceeding via a key *Diels–Alder* cycloaddition reaction (Figure 2A)—which gave rise to an endo-rich mixture of products consisting of up to eight stereoisomers (isomers thereafter), each of which exists as a racemate [20,21,22].

Historically, the eight individual isomers of NRB have been assigned using arbitrary letter designations (R-Y) (Figure 2B). Isomer separation studies conducted by Mohrbacher et al. in the 1960s—using a combination of fractional crystallisation/chromatography and thermal isomerisation techniques—successfully isolated milligram quantities of each NRB isomer [21]. Despite these very low recoveries, sufficient samples were obtained to evaluate the toxicities of the individual isomers in rats by i.v., and, in some instances, by p.o. More recently, chemical derivatisation/chromatography strategies have allowed for the isolation of endo isomers U, V, W, and Y in higher yields [22].

Toxicological studies conducted in vivo established that only the endo isomers (U, V, W, and Y) were lethal to rats, with isomer V being the most potent (LD_50_ 0.15 mg/kg i.v.; 2.1 mg/kg p.o.) (Figure 2B and Table 1) [3,22,23]. It was also noted that the stereochemistry at the carbinol carbon (C-5) of NRB had a pronounced effect on toxicity: endo isomers bearing a threo configuration at this centre (V and Y) were found to be approximately 10-fold more potent than their erythro counterparts (U and W) (Figure 2B and Table 1).

## 2. Norbormide Pharmacodynamics

### 2.1. The Rodenticide Norbormide: In Vivo Pharmacological Activity and Toxicological Studies

In vivo toxicity of NRB has been investigated by Russell [1] and Roszkowski [2] in more than 30 species, comprising both avians and mammals (including humans). NRB showed a safe profile in most of the species assessed, with only minor toxic effects observed after oral administration of doses >1000 mg/kg, a concentration 20 to 100 times higher than the LD_50_ calculated for the *Rattus* genus. The hamster (*Cricetus auratus*) was the most susceptible species after *Rattus*, with an oral LD_50_ value of 140 mg/kg. Russell [1] profiled NRB in four different strains of rats: *R. norvegicus* (wild Norway rat), *R. rattus* (roof rat), *R. hawaiensis* (Hawaiian rat), and *R. norvegicus domestica* (domestic albino rat) and found that toxicity was significantly affected by the route of administration: i.v. gave rise to an LD_50_ value of 0.64 mg/kg in all 4 species, i.p. between 3.5 and 4.4 mg/kg, and oral gavage between 5.3 and 52 mg/kg, dependent upon species. Though i.v. LD_50_ values were comparable in males and females, significant differences were recorded after oral administration, particularly in Norway rats, reflecting different NRB oral bioavailability between genders [2].

### 2.2. Norbormide Poisoning in Rats: Aspects and Mechanisms Involved

The symptoms that follow the ingestion of lethal doses of NRB start approximately 10 min post consumption and are characterised by behavioural changes such as increased motor activity, incoordination and weakening of the hind limbs, blanching of the peripheral extremities and ears, cooling of the body, and laboured breathing. These signs are often associated with anoxic convulsions that precede death. The time to death can be extremely variable, extending from only 15 min to a few days post-dosing: in most cases, however, death occurs between 3 and 4 h post-dosing [1,2,24]. Elevated blood glucose levels have also been reported in NRB-treated rats but not in NRB-treated mice [25].

Several pieces of evidence support the hypothesis that NRBs’ lethal effect in rats is a consequence of diffuse vasoconstriction of the peripheral arteries and veins [2,26]. For example, (1) NRB does not contract the blood vessels of non-*Rattus* species [19]; (2) the symptoms of NRB intoxication are consistent with such a mechanism [2]; (3) among NRB derivatives, only those capable of stimulating rat-isolated arteries in vitro are lethal [3,22]; and (4) among the 8 isomers of NRB, only those endowed with vasoconstrictor activity (i.e., the four endo isomers but not the four exo isomers) are toxic [3].

The chain of events linking vasoconstriction to death, however, remains unclear. According to Roszkowski, microvasculature constriction induced by NRB results in a diffused ischemia that compromises the physiological function of several vital organs and tissues, thus leading to death [2]. This hypothesis, however, contrasts with several experimental indications. First, i.v. administration to rats of 0.5–1.0 mg NRB induces only a transient and weak increase in blood pressure, rapidly followed by a marked hypotensive response with death occurring within 1 min: this period is not sufficient to produce multi-organ damage, though it could potentially induce acute heart failure. Second, in rats killed by oral ingestion of NRB, histological examination of various organs, including the myocardium, revealed only marginal alterations that were not consistent with acute organ failure [27,28]. Third, a diffuse microvasculature constriction is expected to trigger a marked and sustained blood pressure rise rather than hypotension. Nonetheless, the role of hypertension in the lethal effect of NRB cannot be ruled out because the above-mentioned studies [2,29] were performed in anaesthetised rats, and the anaesthetic could have significantly blunted the increase in blood pressure. Therefore, only the monitoring of blood pressure in conscious rats treated with NRB will clarify whether hypertension plays a role in its lethal effect.

A more recent proposal put forward by the authors of this review suggests that NRB may exert its lethal action by the induction of a cardiogenic shock, resulting from diffuse coronary artery constriction, myocardial ischemia, acute heart failure, and hypotension (Figure 3).

This hypothesis is supported by the following observations: (1) the symptoms of NRB intoxication are very similar to those reported in patients with cardiogenic shock [30]; and (2) cardiogenic shock has been reported in patients with normal coronary arteries as a consequence of an acute multivessel coronary spasm [31,32,33], an effect that is also induced by NRB in isolated and perfused rat hearts [2] but not in guinea pig hearts [34] (see below for details). Whether cardiogenic shock plays a key role in NRB-induced lethality is currently under investigation.

### 2.3. In Vitro and Ex Vivo Vascular Effects of Norbormide

Evidence that NRB could induce arterial vasoconstriction was first published by Roszkowski [2], who found that administration of 2–10 µg NRB into the coronary vessels of isolated and perfused rat hearts markedly reduced or even stopped coronary flow. Similar results were obtained when NRB was directly injected in in situ rat mesenteric and ear arteries. However, NRB did not contract rat aortic strips, suggesting that large arteries were not sensitive to the compound [2]. Subsequently, Bova et al. [19] confirmed and extended these results, showing that NRB (0.5–50 µM) induced a concentration-dependent contractile effect in isolated rings from rat caudal, renal, and mesenteric arteries, but not in rat and guinea-pig aorta, guinea-pig mesenteric artery, mouse caudal artery, and human subcutaneous arteries, leading to the conclusion that NRB behaves as a highly selective vasoconstrictor agent for the small arteries of rats (Table 2) [19].

An in-depth characterisation of the vasoconstrictor effects of NRB along the vascular tree of the rat performed using rat arteries and veins of different calibre from various anatomic areas showed that NRB-induced contraction was similar for both arteries and veins but was inversely related to the calibre of the vessel, although veins were more sensitive to NRB than the corresponding arteries (Table 2) [30].

Interestingly, NRB activity in rat arteries was found to be more complex than was initially thought, simultaneously activating both contractile and relaxant mechanisms in the same vessel (Table 2). The activation of the relaxing mechanism can easily be evidenced in rat arteries which are poorly or not contracted by NRB (e.g., rat aorta) or even in non-rat arteries, where precontraction of these vessels by high KCl concentrations is readily reversed by NRB [19,26]; in addition, electrophysiological data have shown that the relaxant mechanism can also take place during NRB-induced contraction [35]. Finally, in NRB-contracted arteries, relaxation can be induced by both the endo (contractile) and the exo (non-contractile) isomers of NRB, suggesting that different binding sites may be involved in regulating relaxation versus contraction [22].

### 2.4. Mechanisms Involved in Norbormide-Induced Vasoconstriction

In vascular myocytes, NRB has been initially proposed to induce vasoconstriction by stimulating a *Rattus*-specific PLC-PKC pathway and by increasing Ca^2+^ influx through both Ca_V_1.2 and store-operated Ca^2+^ channels [19].

PLC activation is a mechanism shared by many endogenous vasoconstrictors (e.g., angiotensin II, endothelin-1, epinephrine, vasopressin, etc.) that bind G protein-coupled receptors on the extracellular side of the plasma membrane [36,37,38]. NRB, however, seems to target an intracellular binding site, as demonstrated by studies employing fluorescent vasoconstrictive NRB derivatives (see below for more details).

An additional mechanism contributing to NRB-induced vasoconstriction has recently been proposed by Saponara et al. [39]. In rat tail-artery myocytes, NRB inhibits K_ATP_ channels that regulate membrane potential: their inhibition causes membrane depolarisation, opening of Ca_V_1.2 channels, and Ca^2+^ entry into the myocyte. This effect, however, was neither stereo-, species-, nor tissue-specific, as both the endo and exo NRB isomers showed comparable K_ATP_ channel inhibitory activity in rat and mouse caudal arteries, as well as in gastric fundus myocytes [39].

Finally, mitochondria may play a role in the NRB contractile effect, as NRB causes a *Rattus*-specific (but not tissue-specific) activation of the mitochondrial permeability transition pore (PTP) that, by inducing the release of mitochondrial Ca^2+^ into the cytoplasm, could potentially contribute to vascular contraction [40] (see below for details) (Figure 4).

### 2.5. Mechanisms Involved in Norbormide-Induced Vasorelaxation

Studies in rat and non-rat artery rings and single myocytes suggest that NRB-induced relaxation is mediated by a reduction of Ca^2+^ entry through Ca_V_1.2 channels (Figure 4). This is supported by several observations: (1) NRB induces a concentration-dependent relaxation of rat aorta rings and of non-rat artery rings that have been stimulated with high KCl concentrations [19], which trigger membrane depolarisation and Ca_V_1.2 channel opening [41]; (2) NRB does not affect phenylephrine-induced contraction [19], where Ca_V_1.2 channels play a minor role [42]; (3) in A7r5 cells derived from the rat aorta, NRB inhibits intracellular Ca^2+^ transients evoked by high KCl concentrations but not by serotonin (the latter is mainly due to the release of Ca^2+^ from the sarcoplasmic reticulum [19]); and (4) in patch-clamp experiments using rat tail-artery myocytes, NRB inhibited Ca_V_1.2 channel currents in a concentration- and voltage-dependent manner, preferentially acting on the inactivated and/or open state of the channel [35]. This contrasts with previously observations from the same preparation, though under different experimental conditions [19], where NRB promoted Ca^2+^ influx through the same Ca_V_1.2 channels. Finally, under the patch-clamp conditions used (i.e., whole-cell configuration causing cytoplasmic dialysis), NRB did not trigger cell contraction, which was consistent with the hypothesis that a cytoplasmic pathway/signalling cascade is required to mediate the vasoconstricting activity of the compound [35].

### 2.6. Effects of Norbormide on Non-Vascular Smooth Muscles

Roszkowski was the first to observe that NRB was unable to contract rat non-vascular smooth muscles in vitro [2]. However, more recent studies performed in isolated rat urinary bladder, tracheal, and duodenal rings showed that rather than being totally ineffective on non-vascular tissues, NRB displayed relaxant activity as a consequence of the inhibition of Ca^2+^ influx through Ca_V_1.2 channels [43]; this activity was similar to that observed previously in rat aorta and non-rat arteries (Table 2) [19].

### 2.7. In Vivo, Ex Vivo, and In Vitro Effects of Norbormide on Rat Heart

The in vivo cardiac effects of NRB have been evaluated by ECG recordings in anaesthetised rats. I.v. administration of the compound caused rapid death of the animal within 15–60 s after injection. During this period, bradycardia and seemingly reversible “arrhythmic episodes” were observed [2]. Significant ECG changes included ST-segment elevation (indicating ischemic myocardial damage), and the appearance of sporadic premature ventricular contractions and, occasionally, ventricular fibrillation [29] was also apparent. However, bradycardia frequently occurs as a consequence of compensatory pauses to premature ventricular contractions, so in anaesthetised rats arrhythmias may not contribute to the lethal effect of i.v. administered NRB. Whether this also pertains to conscious rats after consumption of NRB remains to be elucidated.

More exhaustive and detailed information about the effects of NRB on the mechanical and electrical activity of the heart and coronary circulation has been reported in isolated and perfused rat hearts (Langendorff model). Administration of NRB into the coronary vessels markedly reduced or even stopped both coronary blood flow and inotropic activity, resulting in decreased heart rate, arrhythmias and, in some cases, heart standstill. However, since NRB did not affect the mechanical activity of isolated myocardial strips, only coronary vasoconstriction was considered responsible for the cardiac toxicity [2].

### 2.8. In Vivo, Ex Vivo and In Vitro Effects of Norbormide on Non-Rat Heart

In guinea pigs, the mechanistic action is somewhat different to that observed in rats, in that NRB lowers both the basal coronary pressure and ventricular contractility. It also reduces the atrioventricular conduction of isolated and perfused hearts and decreases the frequency of the sinus atrial node in spontaneously beating right atria. Moreover, in isolated ventricular myocytes, it inhibits the peak Ca^2+^ current through Ca_V_1.2 channels. Qualitatively similar results were obtained with verapamil, a well-known Ca_V_1.2 channel blocker used in therapy for the treatment of hypertension and cardiac arrhythmias [34,44,45]. Taken together, these results indicate that in non-rat hearts, NRB shows a pharmacodynamic profile consistent with the blockade of Ca_V_1.2 channels, analogous to what has been observed in rat aorta, in non-rat arteries, and in rat and non-rat non-vascular smooth muscles (see above). Whether NRB blockade of Ca_V_1.2 channels also takes place in rat hearts, and its possible role in the lethal effect, have yet to be elucidated.

### 2.9. Norbormide and Mitochondria

In 1977, a species-selective action of NRB on mitochondrial function was reported for the first time by Patil and Radhakrishnamurty [46]. In rat liver preparations, low concentrations of NRB-induced mitochondrial swelling, affected ATPase activity, and reduced both succinate- and ATP-driven Ca^2+^ uptake; no changes, however, were observed in mouse preparations [46]. More recent studies have focused on the mitochondrial permeability transition pore (PTP), a Ca^2+^-activated channel located in the inner membrane that has been associated with mitochondrial swelling and cell death pathways [47,48]. NRB induces *Rattus*-selective mitochondrial swelling by opening the PTP through the modification of the Ca^2+^ binding sites of the inner membrane that regulate its function [40]. A more in-depth analysis revealed that *Rattus* selectivity of NRB toward the PTP is not due to a different PTP structure/target, but rather involves a transport system that allows selective uptake of the drug to the inner membrane/matrix of mitochondria. Data suggest that this transport system is the 18 kDa translocator protein TSPO (formerly known as the peripheral benzodiazepine receptor) or a TSPO-associated protein that is unique to the rat [40]. Notably, this NRB-induced PTP activation was neither tissue-specific—as it also occurred in mitochondria isolated from different rat organs (liver, kidney, and heart)—nor stereo-specific, as it was triggered by both endo (lethal) and exo (inactive) isomers. In rats, therefore, NRB mitochondrial effects and lethality seem to be two distinct phenomena [48].

### 2.10. Norbormide and the Adrenal Gland

Several endogenous vasoconstrictors, such as angiotensin-II and endothelin-1, which activate PLC-coupled receptors, also stimulate aldosterone production from the adrenal gland by binding to the same receptor subtypes involved in vascular contraction, namely AT1 and ETA [49,50]. As a PLC-coupled receptor was initially proposed as the target that mediates NRB-induced vasoconstriction [19], the effects of NRB on aldosterone production from the adrenal gland were also investigated. In in vitro rat adrenal zona glomerulosa, NRB increased aldosterone secretion by enhancing the late steps of steroid synthesis responsible for the transformation of deoxycorticosterone to corticosterone and aldosterone, without affecting the early phases of the biosynthesis pathway [51], further supporting the hypothesis that a PLC-coupled receptor is involved in its vasospastic effect.

## 3. Norbormide Pharmacokinetics

To date, only limited studies have focused on elucidating NRB pharmacokinetics, and there is a notable lack of data on its metabolism and distribution.

Metabolic variability across species has long been recognised as a contributing factor to the different pharmacological and toxicological effects of various drugs [52]. Soon after the discovery that NRB was selectively toxic to *Rattus*, experiments were performed to verify if pharmacokinetic differences between species, i.e., oral bioavailability and/or hepatic inactivation, could account for the lack of toxicity in non-rat species. Reduced bioavailability was quickly discounted, as NRB was found to be inactive when administered parenterally in other species [2]. Likewise, rapid hepatic detoxification of NRB was also ruled out as hepatectomised mice receiving a dose of NRB 10 times the rat LD_50_ survived and lived almost as long as control hepatectomised animals [2].

More recently, two studies investigated the role of liver metabolism in the toxic effect of NRB in rats [53]. The metabolic profiling of NRB endo isomers (V, Y, U, and W) and exo isomer X, performed in rat, mouse, and guinea-pig liver preparations, indicated that endo (lethal) isomers were metabolised in both rat and non-rat species through an enzymatic process leading to the production of hydroxylated metabolites. The metabolic process took place primarily in the liver S9 fraction, occurred at very low basal levels in the cytosolic fractions, and was absent in the microsomal fractions. However, no metabolites were detected with the exo (inactive) isomer X. The metabolic profile of the guinea pig was qualitatively similar to that of the rat, though metabolite production was 10 times lower. In contrast, both quantitative and qualitative differences were observed in the mouse. In all the species investigated, metabolite production was significantly higher in females than in males.

NRB metabolism and distribution were also analysed in vivo in rats and mice after oral administration, and in rats after i.v. injection [53]. In orally treated rats, NRB and its hydroxylated metabolites, which were identical to those identified in rat liver preparations [53], were detected in the liver, red blood cells, and heart, but not in plasma, suggesting a very rapid clearance of the compound from the bloodstream. Moreover, an unidentified metabolite (called M3), which was not found in the in vitro liver hepatic S9 fraction, was detected in plasma but not in the other organs. Consistent with the in vitro study, hydroxylated and M3 metabolite production was higher in female compared to male rats. Finally, no metabolites were detected in rats treated with NRB after i.v. injection. In orally treated mice, contrary to rats, unmodified NRB was detected only in plasma 30 min after ingestion; neither the hydroxylated nor the M3 metabolites were found in plasma or in the other organs investigated.

Taken together, these observations highlight the differences in NRB metabolism between rat and non-rat species, which could potentially account for its *Rattus*-selective toxicity. However, the observations that i.v. injection of NRB in rats caused rapid death in the absence of metabolite formation, and the fact that NRB could be detected in mouse plasma in the absence of toxic effects, strongly suggests that different pharmacodynamic features across species are responsible for its species-selective toxicity, and that metabolism plays a role only in the differential susceptibility seen between female and male rats following oral treatment [53].

## 4. Norbormide Analogues/Derivatives and Their Applications

Several NRB analogues/derivatives have been synthesised with the following aims: (1) characterising the pharmacophore responsible for the parent molecules’ vasoconstricting and lethal activities; (2) improving its rodenticidal performance; and (3) identifying its binding sites in the target cells.

### 4.1. Characterisation of the Norbormide Pharmacophore: Norbormide Analogues/Derivatives and Structure–Activity Studies

Via the preparation of a series of “endo-predominant” NRB analogues, studies revealed toxicity towards rats to be highly sensitive to structural changes, with all reported modifications to the aryl/heteroaryl rings (of NRB) resulting in a complete loss of toxicity (Figure 5; modifications relative to NRB are highlighted in green). Conversely, in some instances, substitution at the dicarboximide nitrogen of NRB was tolerated, with several *N*-substituted NRB derivatives being demonstrated to exhibit broadly comparable toxicity to that of the parent molecule; in other examples, marked decreases in LD_50_ were observed (Figure 5; modifications relative to NRB are highlighted in grey) [3,21].

More recently—as part of a “pseudo-deconstruction” approach—a focused series of “truncated” NRB analogues were prepared in an effort to shed further light on the structural features responsible for the parent molecules’ biological profile (Figure 6) [54]. Of the truncated analogues investigated, only compound **3** was found to retain contractile activity in rat caudal artery (Figure 6 and Table 3)—though further experimentation later confirmed the origin of vasoconstriction to be via a different mechanism, since its vasoconstrictor effect could be antagonised by prazosin, a selective alpha-1 adrenergic receptor antagonist—thereby confirming the importance of all three groups (C-5, C-7, and the dicarboximide) for “NRB-type” vasoconstriction. In terms of relaxing effects, of the truncated analogues, only compound **2** was found to be a vasodilator—as demonstrated by its capacity to relax rat aorta pre-contracted by KCl (Table 3)—indicating the groups at C-5 and C-7 (in NRB), but not the dicarboximide, are integral to this response.

Concerning mitochondrial dysfunction, truncated analogues **2** and **3** were found to activate PTP in rat liver mitochondria, at levels comparable to NRB (Figure 6 and Table 3)—highlighting the 2-pyridylbenzylidene subunit at C-7 (in NRB) as a key structural feature for PTP-activation. All truncated analogues were subsequently demonstrated to be non-lethal to rats (40 mg/kg, i.p.) (Table 3)—again restating the fundamental importance of the combined effect of the groups at C-5, C-7, and the dicarboximide for NRBs’ lethal action. Informatively, compound **5**—a NRB analogue where the pyridine ring at C-5 had been switched out for a phenyl ring—was revealed to exhibit a very similar biological profile to NRB, both in vitro (vasoconstrictor activity in rat caudal artery) and in vivo (lethal to rats at 40 mg/kg, i.p.) (Table 3) [54].

### 4.2. Improving NRBs’ Rodenticidal Performance in the Field: NRB Prodrugs

To delay the onset of symptoms and circumvent the bait shyness problem (see above), numerous prodrugs of NRB have been developed. In cage trials, bait containing endo NRB prodrug **6** (Figure 7) was demonstrated to be more palatable to Sprague Dawley rats than bait containing unmodified NRB [16]. Moreover, 100% mortality was recorded (n = 12) versus 25% for unmodified NRB (n = 12). Mortality rates for feral Norway rats consuming bait containing endo NRB prodrug **6** were found to be comparably high (93%, n = 15), but lower mortality in feral ship rats (58%, n = 12) was observed.

### 4.3. Identifying Norbormide Binding Sites in the Target Cells: Fluorescent Norbormide Conjugates

NRB represents an attractive molecule due to its intriguing pharmacological and toxicological features. In particular, the potential involvement of a receptor target that is either exclusively expressed in rat peripheral artery, or one that is part of a family expressed in the blood vessels of all animal species, but which has species- and tissue-selective variants is, without doubt, an attractive hypothesis. In an attempt to identify the cellular binding site implicated in the vasoconstrictor effect of NRB, a series of fluorescently tagged endo NRB derivatives were synthesised and visualised in living cells (both NRB-sensitive and -insensitive) using confocal microscopy [55,56,57]. Critically, these derivatives were first demonstrated to retain the requisite key pharmacological properties of the unlabelled parent molecule, as confirmed by both a retention of vasocontractile response in vitro (rat caudal artery rings) and a lethal endpoint in rats (5 mg/kg, i.v.).

Freshly isolated rat tail-artery myocytes have been used as a model of NRB-sensitive cells, whereas LX2 cells have mostly been used as a model of NRB-insensitive cells. Fluorescent NRB derivatives (Figure 8) were obtained by conjugating endo NRB to either a nitrobenzoxadiazole group (e.g., NRB-AF12 and endo-NRB-NBD-bPA) or a boron dipyrromethene difluoride group (e.g., NRB-MC009 and NRB-ZLW0047) [55,56,57].

In freshly isolated rat tail-artery myocytes, NRB-AF12 showed a rapid internalisation and localisation to intracellular structures, such as the sarcoplasmic reticulum and mitochondria, and the plasma membrane, whereas no fluorescence was observed in the cytoplasm [55] (Figure 9). The same results were obtained with NRB-MC009 (green fluorescence) and NRB-ZLW0047 (red fluorescence), indicating that the distribution of the fluorescent derivatives of NRB was not affected by the fluorescent tag. This finding, together with the observation that both NRB-AF12 and MC009 retain the capacity of the parent compound to induce contraction in rat caudal artery rings, supports the hypothesis that the distribution of the fluorescent NRB-conjugates mirrors that of NRB itself [57]. In particular, co-localisation studies employing NRB-MC009 together with the Ca^2+^ indicator X-Rhod-1^TM^ revealed that NRB-MC009 labelled Ca^2+^-containing intracellular and subsarcolemmal structures, indicating that the plasma membrane fluorescence observed with the fluorescent NRBs was not due to the binding of these compounds to the sarcolemma. This conclusion was supported by the lack of co-localisation of NRB-MC009 with CellMask^TM^, a fluorescent probe used to label plasma membrane [55], and by studies showing that plasma membrane fluorescence was not apparent when freshly isolated rat tail-artery myocytes were exposed to a cell-impermeable, biologically active, fluorescent derivative of NRB (endo-NRB-NBD-bPA) (Figure 8) [56]. Importantly, in vascular myocytes freshly isolated from mice caudal artery, a vessel that is not responsive to NRB, the distribution of NRB-MC009 was the same as that reported in freshly isolated rat tail-artery myocytes [55]. Taken together, these results point to an intracellular binding site for NRB in vascular smooth muscle cells that, when targeted by the drug, can activate the contractile process exclusively in the rat arteries [56].

In LX2 cells, both NRB-AF12 and NRB-MC009 showed an overlapping intracellular distribution, although clearer and longer-lasting fluorescence was obtained with NRB-MC009, potentially due to the superior photo-physical properties of BODIPY over its NBD equivalents [56,58,59].

NRB-MC009 was rapidly internalised by the cells and allowed for clear visualisation of several intracytoplasmic structures and organelles such as the endoplasmic reticulum network, mitochondria, Golgi apparatus, endosomes, and lysosomes [57]. NRB-MC009 did not penetrate the nucleus and did not label the plasma membrane.

Importantly, cytotoxic tests demonstrated that the NRB-MC009 was not toxic to cells, even after long-lasting exposure at concentrations 20 times higher than those required to visualise intracellular structures. Based on these results, NRB-MC009 was proposed as a high-performing fluorescent probe for the labelling of intracellular structures in living cells [56].

In living larvae of *Drosophila Melanogaster*, NRB-MC009 was rapidly internalised by most tissues (except the central nervous system) and showed a preferential localisation to the mitochondria and endoplasmic reticulum [60]. Fluorescent NRB conjugates were tested to verify if they could highlight phenotypic modifications of mitochondria and endoplasmic reticulum in pathological models of *Drosophila*, namely Charcot–Marie–Tooth disease and hereditary spastic paraplegia. In the former, fluorescent NRBs (NRB-MC009 and NRB ZLW0047) highlighted the fragmentation and clustering of the mitochondria in neuronal cell bodies, the disorganisation of the sarcomeric location of mitochondria in the larval muscles, and the clumping around the nuclei, all of which are characteristic phenotypes of this disorder [60]. The fluorescent images were comparable to those obtained using the mitochondrial marker UAS-Mito-GFP in the same model [61]. In the latter, larval muscles labelled with NRB-MC009 and NRB ZLW0047 revealed alterations of the endoplasmic reticulum network [60] comparable to those obtained using UAS-KDEL-GFP in the same model [62]. Consequently, fluorescent NRBs may represent useful tools for live imaging of morphological modifications of mitochondrial and endoplasmic reticulum networks in pathological models developed in *Drosophila* [60,63]. In addition, due to their capacity to stain the gut and its epithelial cells, NRB fluorescent conjugates have also been proposed as useful tools to identify abnormal gut morphology and functionality in both larvae and adult *Drosophila*, and for monitoring food intake and chronic feeding during development or screening assays [60].

## 5. Potential Therapeutic Areas

To our knowledge, only one study has reported a potential therapeutic application for NRB. Nude rats bearing peritoneal metastases were dosed with 17 µCi 5-fluro[2-^14^C]uracil, a known anti-tumour drug, to assess its uptake under different experimental conditions [64]. Noticeably, in rats pre-treated with NRB used as a local vasoconstrictor at a dose of 2.5 mg/kg (U/V isomers) i.p., [2], to help reduce peritoneal resorption of the anti-tumour agent, the drug uptake was increased compared to that in the control group, particularly in large tumours. Although NRB is not suitable for use in humans, the authors concluded that peritoneal vasoconstriction may be a valid strategy for increasing 5-fluorouracil uptake in peritoneal metastases.

## 6. Toxicology and Safety Considerations

### 6.1. Environmental Safety Considerations

Though its use as a rodenticide was terminated due to low palatability and bait shyness, NRB may have entered the environment as a soil contaminant. However, NRB’s physical properties limit its volatilisation from wet or dry soil and suggest that any contaminating water should transition into suspended solids or sediments, potentially at risk of bioaccumulation [65].

### 6.2. Side Effects

Information on toxicity to aquatic organisms is lacking. Mice, humans, and other animal species exposed to oral, dermal, or inhalation routes are only mildly affected by this compound. A case of acute skin inflammation along with the clinical picture of chronic dermatosis was reported in a woman who poured a solution of NRB over her hands, right crus, and left forearm [66]. In men, very high doses (20–300 mg) may cause either vasodilation or vasoconstriction, leading to flushing and a subsequent, minimal fall in systolic blood pressure, which returns to control levels within 2 h [65]. No hyperglycaemic effects were observed in humans. Furthermore, no immune toxicity, genotoxicity or carcinogenicity have been reported.

## 7. Conclusions and Future Perspectives

NRB is a 60-year-old compound that has begun to attract renewed attention in the last three decades as studies working to identify the molecular mechanism/s involved in its unique pharmaco-toxicological profile have ramped up. As a rodenticide, NRB represents an outstanding example of an eco-sustainable compound that aligns with current and future biodiversity goals and that, in the new formulations being developed, has significantly enhanced performance compared to the parent molecule. At the same time, NRB is also an extremely desirable compound from a biological perspective, as it can be viewed as a potential lead compound for use in developing species- and vascular-selective agents that can induce death, at relatively low doses, in many other target pest species. The identification of NRB molecular target(s) is critical in establishing whether it is exclusively expressed in rats or whether it can also be found in other species, as variant isoforms or orthologues. If so, this may lead to the discovery of new species- and vascular-selective mechanisms that are involved in the modulation of vessel tone. Research is currently ongoing with the aim of answering these questions and disclosing “the enigma of NRB”.

## Figures and Tables

**Figure 1 cells-13-00788-f001:**
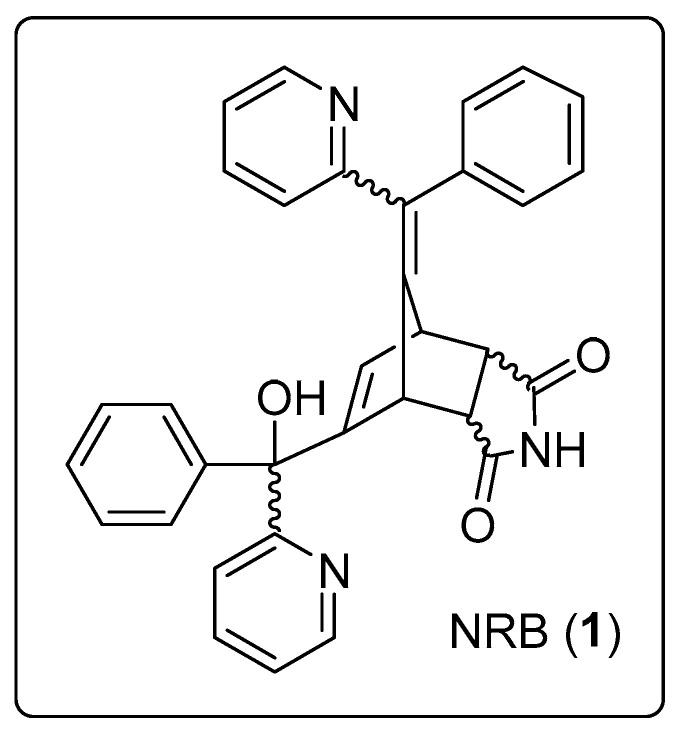
Chemical structure of norbormide (NRB, **1**).

**Figure 2 cells-13-00788-f002:**
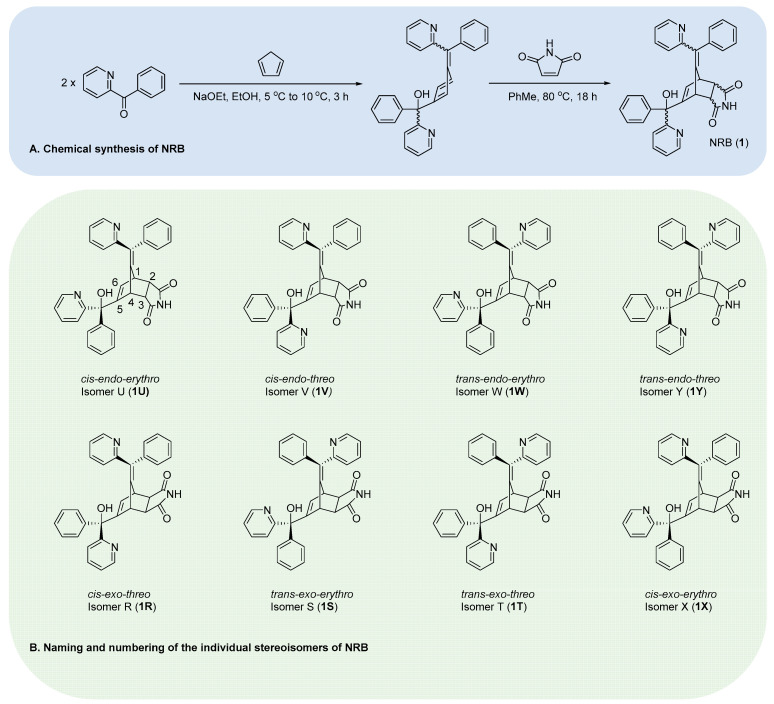
(**A**) Chemical synthesis of NRB (**1**) and (**B**) naming/numbering of the individual stereoisomers of NRB (**1R**-**Y**).

**Figure 3 cells-13-00788-f003:**
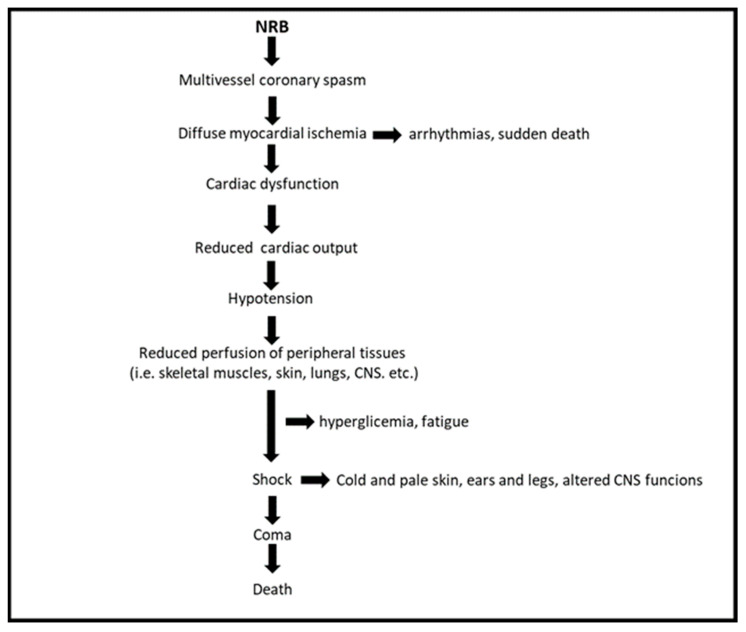
Proposed mechanism for the lethal action of NRB.

**Figure 4 cells-13-00788-f004:**
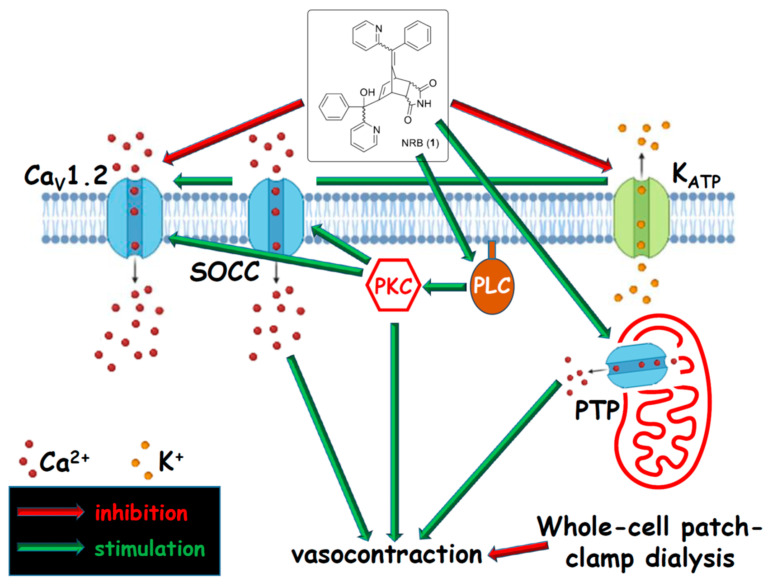
Possible pathways, ion channels, and enzymes, targeted by norbormide, underpinning its vasorelaxant and vasoconstricting activities in rat vascular myocytes. Ca_V_1.2—Ca_V_1.2 channel; K_ATP_—ATP-dependent K^+^ channel; SOCC—store-operated Ca^2+^ channel; PKC—protein kinase C; PLC—phospholipase C; PTP—permeability transition pore. For further details see text.

**Figure 5 cells-13-00788-f005:**
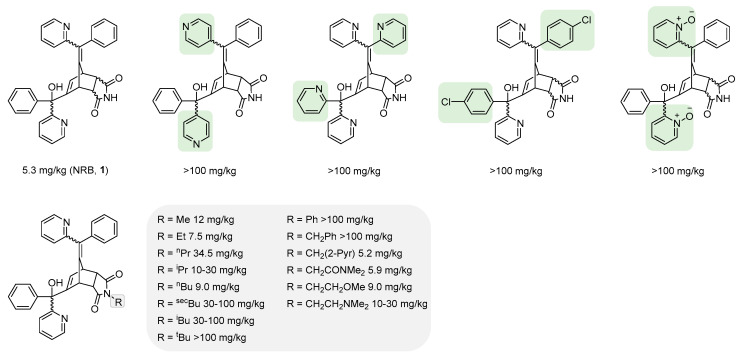
Selected examples of known NRB analogues/derivatives including LD_50_ in rats (p.o.).

**Figure 6 cells-13-00788-f006:**
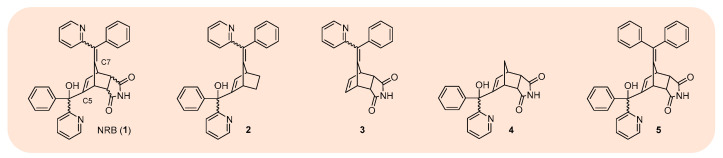
NRB (**1**) and selected examples of “truncated/modified” NRB analogues (**2**–**5**).

**Figure 7 cells-13-00788-f007:**
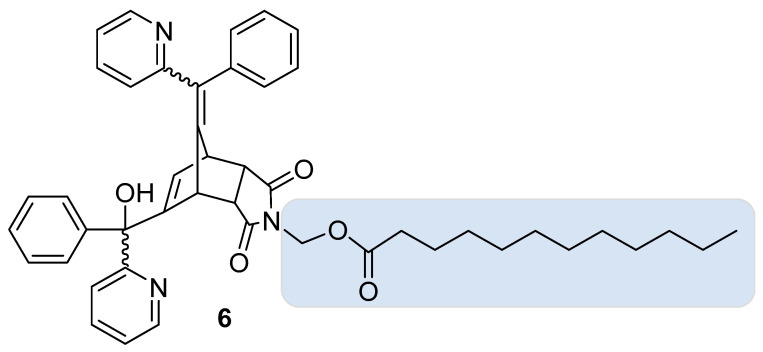
A selected example of a NRB prodrug (**6**).

**Figure 8 cells-13-00788-f008:**
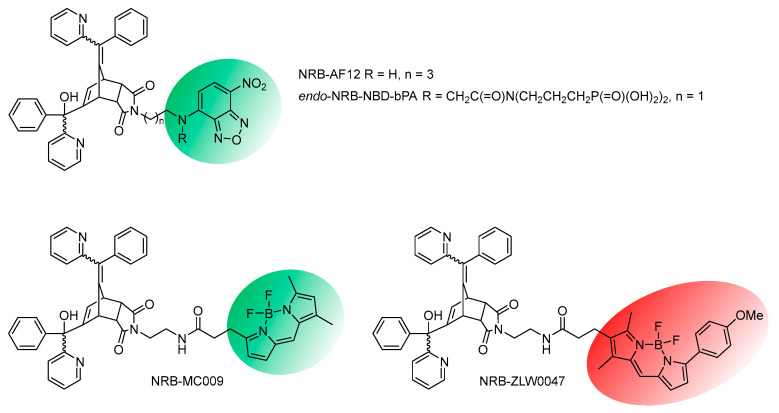
Selected examples of known fluorescent NRB conjugates.

**Figure 9 cells-13-00788-f009:**
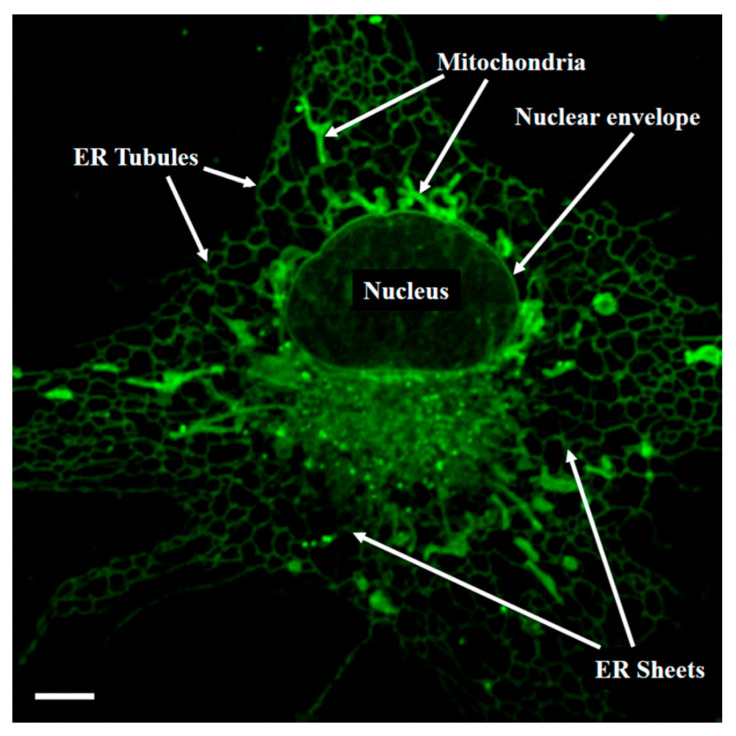
Confocal live imaging of an LX2 cell stained with NRB-AF12 showing the subcellular distribution of the fluorescent probe. Magnification 100×; scalebar 5 μm (from [55]).

**Table 1 cells-13-00788-t001:** In vivo toxicity of NRB (**1**) and NRB isomers (**1R**-**Y**) in rats.

	LD_50_ (Rat) mg/kg	
Compound	i.v.	p.o.
**1** ^a^	0.65	5.3
**1U**	1.25-1.5	N.D.
**1V**	0.15	2.1
**1W**	5.0	N.D.
**1Y**	0.5	<5
**1R**	>10	N.D.
**1S**	>8.0	N.D.
**1T**	>8.5	N.D.
**1X**	>10	>100

^a^ In vivo data collected using different batches of mixed NRB. N.D.: not determined.

**Table 2 cells-13-00788-t002:** Contracting and relaxant effects of norbormide on various smooth muscles.

	Smooth Muscle Type				
	Vascular				Non-Vascular
	Contraction		Relaxation	No Effect	Relaxation
	Artery	Vein	Artery	Artery	
Rat	Coronary	Pulmonary	Thoracic aorta		Urinary bladder
	Mesenteric	Thoracic vena cava	Abdominal aorta		Tracheal
	Ear	Abdominal vena cava	Carotid		Duodenal
	Caudal	Iliac	Pulmonary		
	Renal	Jugular	Iliac		
	Pulmonary				
	Abdominal aorta				
	Iliac				
	Femoral				
Mouse				Caudal	
Guinea pig				Aorta	
				Mesenteric	
Man				Subcutaneous	

**Table 3 cells-13-00788-t003:** Vasoconstriction, vasodilation, mitochondrial permeability transition pore stimulation (PTP opening), cardiotoxicity (coronaro-constriction in isolated and perfused rat hearts), and in vivo lethality of NRB (**1**) and “truncated/modified” NRB analogues **2**–**5**.

Compound	Vasoconstriction (EC_50_) ^a^	Relaxant Activity ^b^	PTP Opening ^c^	Coronaro-Constriction	In Vivo Lethality
**1**	209% (1 µM)	64%	++++	Yes	Yes ^d^
**2**	-	60%	+++	No	No
**3**	193% (7.8 µM)	-	++++	No	No
**4**	-	-	+	No	N.D.
**5**	185% (4.2 µM)	42%	++	Yes	Yes ^e^

^a^ Data are from rat caudal arteries and represent the % of KCl-induced contraction. ^b^ Data are from rat aorta and represent the % of relaxation of KCl-induced contraction. ^c^ Stimulation of PTP opening: + symbols indicate the degree of stimulation. ^d^ 20 mg/kg (i.p., rat)—induced death after 1 h. ^e^ 40 mg/kg (i.p., rat)—induced death after 1 h. N.D.: not determined.

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
