# Peer review of "The Enigma of Norbormide, a Rattus-Selective Toxicant"

_cells, 2024, doi:10.3390/cells13090788_

Round 1

Reviewer 1 Report

Comments and Suggestions for Authors

The manuscript titled “The Mystery of Norbormide, a Rattus-Selective Toxicant” by Fusi et al. documents various aspects of norbormide (NRB), including its chemical and physical properties, pharmacodynamics, toxicology, reported health effects, potential applications, and mechanisms of action. Additionally, the authors also provide information on recent developments regarding NRB derivatives. Overall, it is a well-written and detailed review summarizing NRB and its derivatives’ potential as a rat-selective, synthetic toxicant. Below are several points that require further attention.

1)       Figure 4:  The caption should include the full names for the abbreviations used in the figure.

2)       Page 7 last paragraph:  References are missing.

3)       Page 12 section 4.2:  References are needed.

4)       Page 13 first paragraph:  References are needed.

5)       Page 14 first paragraph:  References are needed.

6)       Figure 9:  Font is too small in the figure labels.

Author Response

Response to Reviewer #1 - Open Review

Comments and Suggestions for Authors

Below are several points that require further attention.

1) Figure 4: The caption should include the full names for the abbreviations used in the figure.

Response. Done.

2) Page 7 last paragraph: References are missing.

Response. Reference has been added (l. 245).

3) Page 12 section 4.2: References are needed.

Response. Reference has been added (l. 455).

4) Page 13 first paragraph: References are needed.

Response. References have been added (l. 471).

5) Page 14 first paragraph:  References are needed.

Response. Reference has been added (l. 514).

6) Figure 9: Font is too small in the figure labels.

Response. The label font has been increased.

Reviewer 2 Report

Comments and Suggestions for Authors

I like this review I have just a few suggestions for authors;

- Can they create a figure with all toxicity studies performed in different studies with all organs?

Do authors need to add more evidence in sentences in line 483? What do they mean by hibiting morphological mitochondria? Mitochondria fusion defects in clumping (Franco A. Life 2022) or mitophagy (Li J. et alFrontiers in cell biology and Development)? Can they try to add more sentences?

Comments on the Quality of English Language

The review can be published after minor revision. 

Author Response

Response to Reviewer #2 - Open Review

Quality of English Language

(x) Minor editing of English language required

Response. Done by the three native English-speaking authors.

Comments and Suggestions for Authors

I like this review I have just a few suggestions for authors.

  1. Can they create a figure with all toxicity studies performed in different studies with all organs?

Response. We thank the Reviewer for this good suggestion. However, as stated in the article (see paragraph starting at line 162), post-mortem evaluation of rats killed by norbormide ingestion revealed non-specific and marginal morphological alterations, that were the same in all the organs investigated (Niu, 1970; De Nicolo, 2002). For this reason, we believe that a Figure showing norbormide-induced organ toxicity would not add value to the review.

  1. Do authors need to add more evidence in sentences in line 483?

Response. Two new sentences have been included in the revised review (l. 489-493).

  1. What do they mean by hibiting morphological mitochondria? Mitochondria fusion defects in clumping (Franco A. Life 2022) or mitophagy (Li J. et all Frontiers in cell biology and Development)? Can they try to add more sentences?

Response. The last paragraph of Section 4.3 has been modified according to the suggestions of the Reviewer (l. 529-546).

Reviewer 3 Report

Comments and Suggestions for Authors

The article titled "Unveiling the Mystery of Norbormide: A Rattus-Selective Toxicant," identified with the reference code cells-2972341 and submitted for review in Cells (ISSN 2073-4409), serves as an insightful review elucidating the development process and implications of Norbormide, along with its novel prodrug formulation acclaimed for its eco-sustainability and Rattus-selective toxicity.

Apart from its historical narrative detailing the discovery process, physical attributes, chemical synthesis, and stereochemistry of Norbormide, the paper delves into its pharmacodynamics and pharmacokinetics, alongside discussions on potential therapeutic applications and considerations regarding toxicology and safety. Abundantly enriched with figures and tables, the paper notably includes visual representations elucidating Norbormide's chemical structure, synthesis process, and stereochemical variants. Moreover, it presents a proposed mechanism elucidating Norbormide's lethal action, as well as potential pathways for its vascular effects. While acknowledging the scarcity of pharmacokinetic data, the authors adeptly extract and synthesize existing literature to provide a comprehensive understanding of Norbormide's pharmacological profile.

The paper admirably encompasses various aspects of Norbormide and its derivatives; however, one area deserving further exploration is their potential in imaging applications. Incorporating this aspect would significantly enhance readers' comprehension of the compound's overall importance. Two relevant articles could be included in the reference list to supplement this discussion.

1. Forgiarini A, Wang Z, Bova S, Brimble MA, Hopkins B, Rennison D, Orso G. Norbormide-Based Probes and Their Application for Mitochondrial Imaging in Drosophila Melanogaster. Methods Mol Biol. 2021;2275:279-289. doi: 10.1007/978-1-0716-1262-0_17. PMID: 34118044.

2. Wang ZL, Li FF, Quach R, Ferrarese A, Forgiarini A, Ferrari M, D'Amore C, Bova S, Orso G, Fusi F, Saponara S, Hopkins B, Brimble MA, Rennison D. Nitrobenzoxadiazole derivatives of the rat selective toxicant norbormide as fluorescent probes for live cell imaging. Bioorg Med Chem. 2022 Apr 1;59:116670. doi: 10.1016/j.bmc.2022.116670. Epub 2022 Feb 17. PMID: 35202967.

The first reference presents positive results regarding the imaging potential of Norbormide-based compounds, while the second reference offers contrasting findings with negative results. Both references are pertinent and directly related to the topic of this review paper, providing valuable insights into the diverse outcomes and implications of Norbormide research. 

Generally, the selected references by authors are not only pertinent but also closely aligned with the topic of this review paper. They effectively complement the discussion by providing substantial insights into the field. The authors' significant contributions to the research domain are evident from the cited references, reflecting their expertise and comprehensive understanding of the subject matter.

Author Response

Response to Reviewer#3 - Open Review

Comments and Suggestions for Authors

  1. The paper admirably encompasses various aspects of Norbormide and its derivatives; however, one area deserving further exploration is their potential in imaging applications. Incorporating this aspect would significantly enhance readers' comprehension of the compound's overall importance.

Response. The last paragraph of Section 4.3 now includes a brief discussion on the potential of NRB derivatives in imaging applications (l. 529-546).

  1. Two relevant articles could be included in the reference list to supplement this discussion.
  2. Forgiarini A, Wang Z, Bova S, Brimble MA, Hopkins B, Rennison D, Orso G. Norbormide-Based Probes and Their Application for Mitochondrial Imaging in Drosophila Melanogaster. Methods Mol Biol. 2021;2275:279-289. doi: 10.1007/978-1-0716-1262-0_17. PMID: 34118044.
  3. Wang ZL, Li FF, Quach R, Ferrarese A, Forgiarini A, Ferrari M, D'Amore C, Bova S, Orso G, Fusi F, Saponara S, Hopkins B, Brimble MA, Rennison D. Nitrobenzoxadiazole derivatives of the rat selective toxicant norbormide as fluorescent probes for live cell imaging. Bioorg Med Chem. 2022 Apr 1;59:116670. doi: 10.1016/j.bmc.2022.116670. Epub 2022 Feb 17. PMID: 35202967.

The first reference presents positive results regarding the imaging potential of Norbormide-based compounds, while the second reference offers contrasting findings with negative results. Both references are pertinent and directly related to the topic of this review paper, providing valuable insights into the diverse outcomes and implications of Norbormide research.

Response. As suggested by the Reviewer, the first reference is now cited in the review (number 63). The second one was already cited (number 56).